# Prevalence of psychiatric disorders in Norwegian 10-14-year-olds: Results from a cross-sectional study

Tormod Bøe[1,2]*, Einar Røshol Heiervang[3,4], Kjell Morten Stormark[2,5], Astri J. Lundervold[6], Mari Hysing[1,2]

1 Faculty of Psychology, Department of psychosocial science, University of Bergen, Bergen, Norway, 2 RKBU West, NORCE Norwegian Research Center AS, Bergen, Norway, 3 Institute of Clinical Medicine, University of Oslo, Oslo, Norway, 4 Oslo University Hospital, Oslo, Norway, 5 Department of health promotion and development, University of Bergen, Bergen, Norway, 6 Department of Biological and Medical Psychology, University of Bergen, Bergen, Norway

* tormod.boe@uib.no

**Data Availability Statement:** Data cannot be shared publicly due to privacy restrictions in accordance with the ethical approval for the Bergen Child Study. Norwegian Health research legislation

## Abstract

Youth mental health problems is the leading cause of disability worldwide and a major public health concern. Prevalence rates are needed for planning preventive interventions and health care services. We here report Norwegian prevalence estimates for youth mental disorders based on findings from the Bergen Child Study cohort. A web-based psychiatric interview; the Development and Well-Being Assessment, was completed by parents and teachers of 2,043 10-14-year-olds from the city of Bergen, Norway. Post-stratification weights were used to account for selective participation related to parental educational in the estimation of prevalence rates. Prevalence rates are presented for the whole sample and stratified by gender and age. The overall population weighted estimate suggests that 6.93% (95% CI 5.06–9.41) of the children met DSM-IV diagnostic criteria for one or more psychiatric disorders. There were no robust indications of age- or gender-related differences in the prevalence. 11.4% of the children fulfilled criteria for more than one diagnosis. The most common comorbid conditions were ADHD and disruptive disorders. The prevalence of psychiatric disorders was relatively low among Norwegian 10-14-year-olds, compared to published worldwide prevalence estimates. This is in line with estimates from prior studies from the Nordic countries. These findings raise important questions about the origins of different prevalence rates for psychiatric disorders between societies. The findings also illustrate the importance of locally driven epidemiological studies for planning preventative efforts and appropriately scaling mental health services to meet the need of the population.

## Introduction

Mental health problems among children are a major public health concern and the leading cause of disability worldwide [1]. Further, mental health problems in adulthood often have their origin in childhood and adolescence [2], underscoring the need for reliable prevalence

and the Norwegian Ethics committees require explicit consent from participants in order to transfer health research data outside of Norway. In this specific case, ethics approval is also contingent on storing the research data on secure storage facilities located in our research institution. Data are from the Norwegian Bergen Child Study, owned by NORCE Norwegian Research Centre. The authors did not have special access privileges to data from the survey. Data are available from the the Bergen Child Study Institutional Data Access (contact via bib@norceresearch.no) for researchers who meet the criteria for access to confidential data.

**Funding:** The author(s) received no specific funding for this work.

**Competing interests:** The authors have declared that no competing interests exist.

data. Still, the global coverage of prevalence data for children is limited [3]. The assumed increase in childhood psychiatric disorders over the last decades has been based on official clinical registries [4], but these are susceptible to reporting practices and may therefore not render reliable prevalence estimates [5,6]. Estimates from administrative registers may also be biased since groups with lower socioeconomic status are under-represented [7,8]. Actual prevalence may also be higher because children with mental health problems go undetected and are thus left without treatment options [9,10]. We therefore need population-based epidemiological studies in order to obtain realistic prevalence estimates as a base for offering preventive interventions and scaling of health services.

A meta-analysis of 41 studies from 27 countries produced an overall worldwide prevalence estimate of 13.4% for child mental health disorders, but with large heterogeneity across studies and countries [11]. Differences may result from a number of methodological differences; e.g. sampling procedure, diagnostic instrument, informant type [12], the time frame (three [13], six [12] or 12 months [14]), and the estimation method. However, it may also reflect true variations in prevalence rates a across societies or age groups. Results from a former wave of the present longitudinal study, conducted when the cohort were 7–9 years old, indicated a 7.0% weighted prevalence for psychiatric disorders, with a high rate of comorbid conditions [10]. This was well below the overall world prevalence reported by Polanczyk et al. in their recent meta-analysis [11], but in line with previous studies from Scandinavian countries [15–18].

The rate of psychiatric disorders may also vary by gender and age. While boys show a higher frequency of Attention Deficit Hyperactivity Disorder (ADHD) and disruptive disorders than girls [19–22], there is a predominance of girls having emotional [12,23] and eating disorders [24]. The developmental changes during pre-adolescence may also give rise to changes in rates for mental disorders. It has for example been shown that ADHD is more frequent among younger children whereas conduct and mood disorders are more frequent in early adolescence [23]. Still, many studies present overall estimates for the whole childhood period, not specifying prevalence rates for the different age groups [11,12].

There is also an association between socioeconomic factors and higher frequencies of mental health problems in children and youth [25–27]. Low parental education levels and poorer financial status are also predictors for non-participation in surveys [28]. Thus, if epidemiological studies do not account for selective participation related to higher socioeconomic status (SES), prevalence estimates may be biased [29,30].

The aim of the present study is to present prevalence of psychiatric disorders in a Norwegian population-based sample of early adolescent boys and girls. We hypothesized that 1) estimates would be similar to what was reported for the same cohort at a younger age, 2) that boys would show more externalizing problems and girls more internalizing problems, and 3) an overall increase in the prevalence of emotional and conduct problems in the older age groups.

## Methods

### Participants and procedure

The current analyses are based on data from the Bergen Child Study, a series of cross-sectional multi-phase surveys of children born between 1993 and 1995 living in Bergen, the second largest city in Norway (see https://www.norceresearch.no/en/projects/the-bergen-child-study for more information).

The present study uses data from the second cross-sectional study (wave two) carried out in 2006, when the children were in fifth to seventh grade (10–14 years old), in a target population of 9,218. Mean age was 12.5 (SD = 0.8, range = 10.4–14.2), with 52% of the sample being female. In the first *questionnaire* phase of the data collection, parents, children and teachers

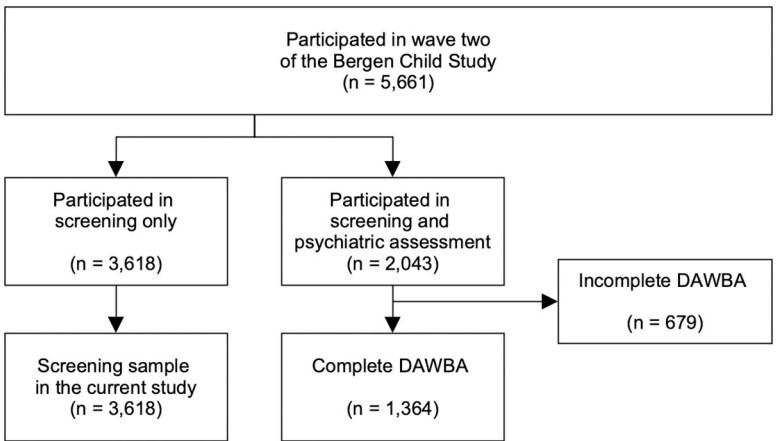

**Fig 1. Participant flow through the two phases of the second wave of the Bergen Child Study.**

completed on-line questionnaires for 5,661 children (teacher data from the screening phase is not included in the following study). Parents who took part in the questionnaire phase were also invited to complete a second phase, which involved an on-line psychiatric interview; the Development and Well-Being Assessment [31], see flowchart in Fig 1.

The DAWBA was administered through a web-based system, and parents and teachers provided information about the children using a special website after logging in with a unique identification number and password. From the 2,043 parents who logged into the on-line interview, 1,364 completed all sections and forms the DAWBA sample in the current study. The study was approved by the Regional Committee for Medical Research Ethics in Western Norway (approval number 062–06) and by the National Data Inspectorate. Participants gave written informed consent and parents and guardians consented on behalf of their children.

### Instruments

**Questionnaire phase.**　Among the instruments included in the questionnaire phase was the Strengths and Difficulties Questionnaire [32,33]. The SDQ is a brief screening questionnaire including 25 items assessing emotional symptoms, conduct problems, hyperactivity-inattention, peer problems and prosocial behaviours. The SDQ total problems score is computed by adding all items except those measuring prosocial behaviours. In the current study, the parent completed SDQ was used to assess the representativeness of participants in the psychiatric assessment phase (i.e. DAWBA sample) compared to those participating in the first phase only.

Parents provided details about their education level (categorized into an ordinal variable with four levels: "Elementary", "High school", "University/college less than or equal to four years" and "University/college more than four years"). Information about child gender and age was obtained from their personal identification number.

**Psychiatric assessment.**　The Development and Well-Being Assessment (DAWBA) is a well-established psychiatric interview [31] and was administered on-line to parents and teachers. The DAWBA is a partially structured interview, containing screening questions with fixed response options, supplemented by open-ended questions if problems are reported. Respondents typed their answers, giving examples and details about level of severity (functional impairment) in text boxes. A highly trained and experienced clinical rater (ERH) subsequently assigned psychiatric diagnoses, with the help of the DAWBA program rater screens [34,35].

More details about the DAWBA is available on the website: http://dawba.info/a0.html. The current study made use of DSM-IV diagnostic categories.

## Statistical analyses

Two prevalence estimates were produced: one unweighted estimate, and one estimate using post-stratification weights. Post-stratification weights were created using auxiliary information about the marginal distribution of education levels in the total population of Norway (c.f. Table 1). Separate weights were created for maternal and paternal education levels. For the population weights, we used information on education levels from Statistics Norway [36]. To account for the fact that the parents reported on 10-14-year-olds (and could possibly also have older children), we used information about education levels for adults aged 30–59. Weights were applied by use of iterative post-stratification (i.e. raking), matching marginal distributions of the survey sample to population margins and extreme weights were trimmed to the median weight plus 5 times the interquartile range of the weights [37,38]. Post-stratification weighting and analyses incorporating the weights were conducted with the R-packages 'survey' [39] and 'srvyr' [40] in R for Mac, version 3.5.3 [41]. Confidence intervals around prevalence estimates were calculated with the *xlogit* method [39]. The association between age, gender and their interaction on the prevalence of different main categories of disorder was tested using survey-weighted generalised linear models in the 'survey' package [39]. Prevalence estimates are reported for broader groups of disorders: *any psychiatric disorders* (all psychiatric diagnoses made), *anxiety disorders*, *depressive disorders*, *ADHD* and *disruptive disorders* (i.e., conduct disorder, oppositional defiant disorder, and other disruptive disorders). Estimates for specific disorders are presented in supplementary tables due to higher uncertainty in these estimates. When estimating age and gender-specific prevalence estimates, age was categorized into five approximately equally sized groups ($n$ = 322–326) corresponding to the age intervals [10.4,11.7), [11.7,12.2), [12.2,12.8), [12.8,13.4) and [13.4,14.2]. To fully investigate the gender-specific influence of weighting on the prevalence estimates, we calculated gender specific correlations between parental education levels and psychiatric disorders, and we also investigated the distribution of psychiatric disorders according to parental education levels on the unweighted data (results available on request).

**Table 1. Distribution of variables used for weighting the survey data.**

|  | DAWBA sample | Population sample |
| --- | --- | --- |
|  | % | % |
| Maternal education |  |  |
| Elementary | 3.7 | 21.4 |
| High school | 31.4 | 41.6 |
| Uni/college (<4 yrs) | 33.1 | 28.9 |
| Uni/college (>4 yrs) | 28.1 | 6.2 |
| Missing | 3.7 | 2.0 |
| Paternal education |  |  |
| Elementary | 7.0 | 20.5 |
| High school | 29.6 | 47.8 |
| Uni/college (<4 yrs) | 26.4 | 19.9 |
| Uni/college (>4 yrs) | 31.2 | 9.5 |
| Missing | 5.9 | 2.2 |

### Representativeness of the DAWBA sample to the screening sample

The representativeness of the participants in the psychiatric assessment phase has been reported in a previous publication, where it was found that those who participated in the DAWBA were slightly younger, had better economic well-being, and had more highly educated parents compared to those participating in the questionnaire phase only [42]. There were minor differences in symptoms of mental health problems between participants in the questionnaire sample and the psychiatric assessment. A Welch two-sample *t*-test yielded significant differences in the means for SDQ Conduct problems and Hyperactivity/inattention, but effect sizes were *very small* [43]. There were no significant differences between the samples regarding the distribution of the SDQ total problems score; 9.5% of participants in the questionnaire sample and 8.4% of participants in the psychiatric assessment sample who scored above the 90th percentile (score >13), $\chi^2(1) = .001$, $p = .969$.

## Results

In the population-weighted analyses, the prevalence of any psychiatric disorder was estimated to be 6.93% (95% CI 5.06–9.41). The most commonly made diagnoses were anxiety disorders, followed by disruptive disorders and ADHD, see Table 2. It was estimated that about 1% the sample had other psychiatric disorders, such as PDD/Autism and Tic disorder (see S1 Table). The estimated point prevalence rates were somewhat higher than in the analyses of the unweighted data.

### Prevalence by gender and age

The prevalence of psychiatric disorders was estimated to be 7.52% (95% CI 4.68–11.86) for girls and 6.27% (95% CI 4.42–8.83) for boys in the gender stratified population-weighted analyses (c.f. Table 3). The most common diagnostic group, as indicated by the point prevalence estimate, was anxiety disorders, followed by disruptive disorders for boys and ADHD for girls. In the regression analyses, there was no evidence of male or female gender significantly predicting any of the major disorder categories (all *p*-values >.2 for anxiety, depression, ADHD, and disruptive disorders). The prevalence rates for all diagnoses are available in S2 Table.

The prevalence by age and gender for the broader disorder categories can be seen in Fig 2. The point prevalence of any psychiatric disorder was highest among the youngest participants, with anxiety disorders, ADHD and disruptive disorders being the most common diagnoses in this age group. Among the oldest participants, the point prevalence is lower, mainly including children with anxiety disorders and ADHD. Anxiety disorders were diagnosed in all age

**Table 2. Prevalence of main categories of DSM-IV disorders among 10-14-year-old participants in the Bergen Child Study.**

|  | Unweighted[1] | | Weighted to population[2] | |
| --- | --- | --- | --- | --- |
|  | % (n) | 95% CI | % | 95% CI |
| Any psychiatric disorder | 5.87 (80) | 4.73–7.25 | 6.93 | 5.06–9.41 |
| Any anxiety disorder | 2.86 (39) | 2.10–3.89 | 3.84 | 2.50–5.85 |
| Any depressive disorder | 0.44 (6) | 0.20–0.98 | 0.64 | 0.16–2.58 |
| Any ADHD | 1.17 (16) | 0.72–1.91 | 1.54 | 0.76–3.10 |
| Any disruptive disorder | 1.32 (18) | 0.83–2.09 | 1.62 | 0.82–3.18 |

[1]DAWBA participants only.
[2]Weighted to population margins for education levels in the population.

**Table 3. Prevalence of DSM-IV disorders among 10-14-year-old participants in the Bergen Child Study, by gender.**

| | Unweighted[1] | | | | Weighted to population[2] | | | |
|---|---|---|---|---|---|---|---|---|
| | Girls | | Boys | | Girls | | Boys | |
| | % | 95% CI | % | 95% CI | % | 95% CI | % | 95% CI |
| Any psychiatric disorder | 4.64 | 3.32–6.46 | 7.20 | 5.45–9.46 | 7.52 | 4.68–11.86 | 6.27 | 4.42–8.83 |
| Any anxiety disorder | 2.81 | 1.82–4.32 | 2.91 | 1.86–4.52 | 4.37 | 2.35–8.00 | 3.25 | 1.93–5.40 |
| Any depressive disorder | 0.56 | 0.21–1.49 | 0.31 | 0.08–1.22 | 1.13 | 0.25–4.96 | 0.09 | 0.02–0.38 |
| Any ADHD | 0.56 | 0.21–1.49 | 1.84 | 1.04–3.21 | 1.61 | 0.50–5.06 | 1.47 | 0.73–2.91 |
| Any disruptive disorder | 0.56 | 0.21–1.49 | 2.14 | 1.27–3.59 | 1.19 | 0.28–4.91 | 2.10 | 1.11–3.95 |

[1]DAWBA participants only.

[2]Weighted to population margins for education levels in the population.

groups, depression mainly among 12.2–12.8-years-old, ADHD mainly among the youngest and oldest participants, whereas disruptive disorders were diagnosed in all age groups except among the 11.7–12.2-years-old. Due to the relatively few participants diagnosed, these age and gender-specific prevalence estimates were highly influenced by how the age-intervals were created. As indicated by the large and overlapping confidence intervals, age, nor its interaction with gender, did not significantly predict any of the major disorder categories (anxiety, depression, ADHD, and disruptive disorders) in the regression analyses, all $p$-values $> .2$.

For a different perspective on age-of onset and accumulation of psychiatric disorders, we also calculated the cumulative prevalence by age and gender in unweighted data, see Fig 3. The steepness of the slopes indicates at which age diagnoses accumulate. For any psychiatric disorder and anxiety disorders there appears to be a steady increase in diagnosed disorders with age. For anxiety disorders, more girls are diagnosed from around 13 years of age, and the cumulative number of females with anxiety disorders become higher than for males. For depressive disorders diagnoses start to accumulate for girls at around 12 years of age, and at a somewhat later age for boys. For ADHD and disruptive disorders, one accumulation appears between 11 and 12 years of age, and another between 13 and 14 years of age, but girls are diagnosed at a younger age.

## Comorbidity

Within the major domains of mental health problems (emotional disorders [anxiety disorders and depressive disorders], conduct disorders and ADHD), 88.6% fulfilled the diagnostic criteria for a single disorder, and 11.4% of children fulfilled criteria for more than one diagnosis, illustrated with the Venn-diagram in Fig 4. The most common overlap was between conduct disorder and ADHD.

## Discussion

The aim of the current study was to estimate prevalence rates for psychiatric disorders in Norwegian 10-14-year-olds. According to the population weighted interval estimate 5.06–9.41% of participants met DSM-IV criteria for a psychiatric disorder, with emotional disorders being most prevalent, followed by ADHD and conduct disorders. There were no suggestions of strong gender and age effects on the prevalence estimates.

### Prevalence of psychiatric disorders

The point prevalence estimates for any psychiatric disorders obtained in this study, and in the previous study of younger children [10] is relatively low compared to most international

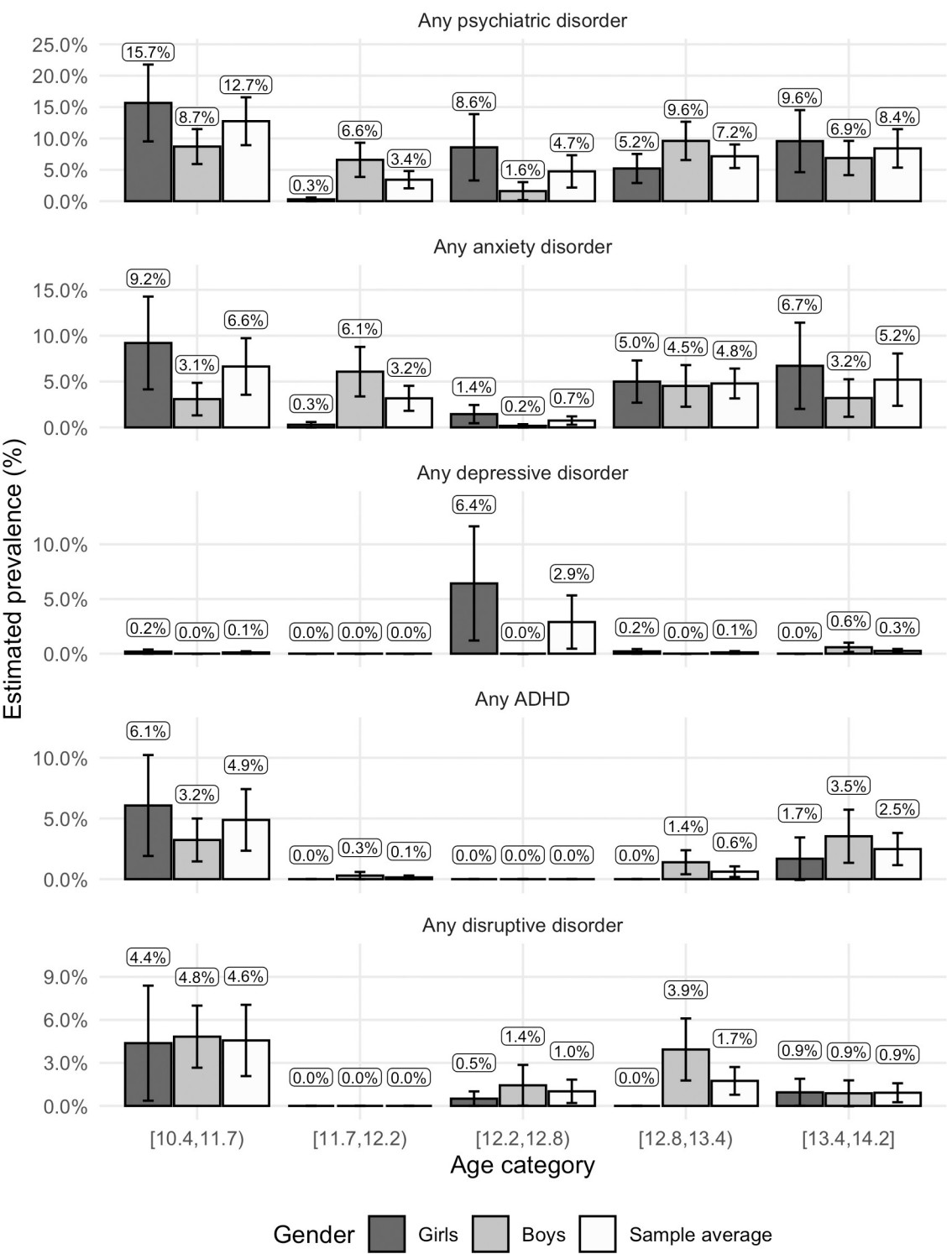

**Fig 2. Prevalence of main psychiatric disorders by age.** Population weighted estimates. Participant age (range 10.4–14.2 years) was categorized into five approximately equally sized intervals corresponding to the categories on the x-axis (n = 322–326). Error bars represent the 95% confidence intervals of the point prevalence.

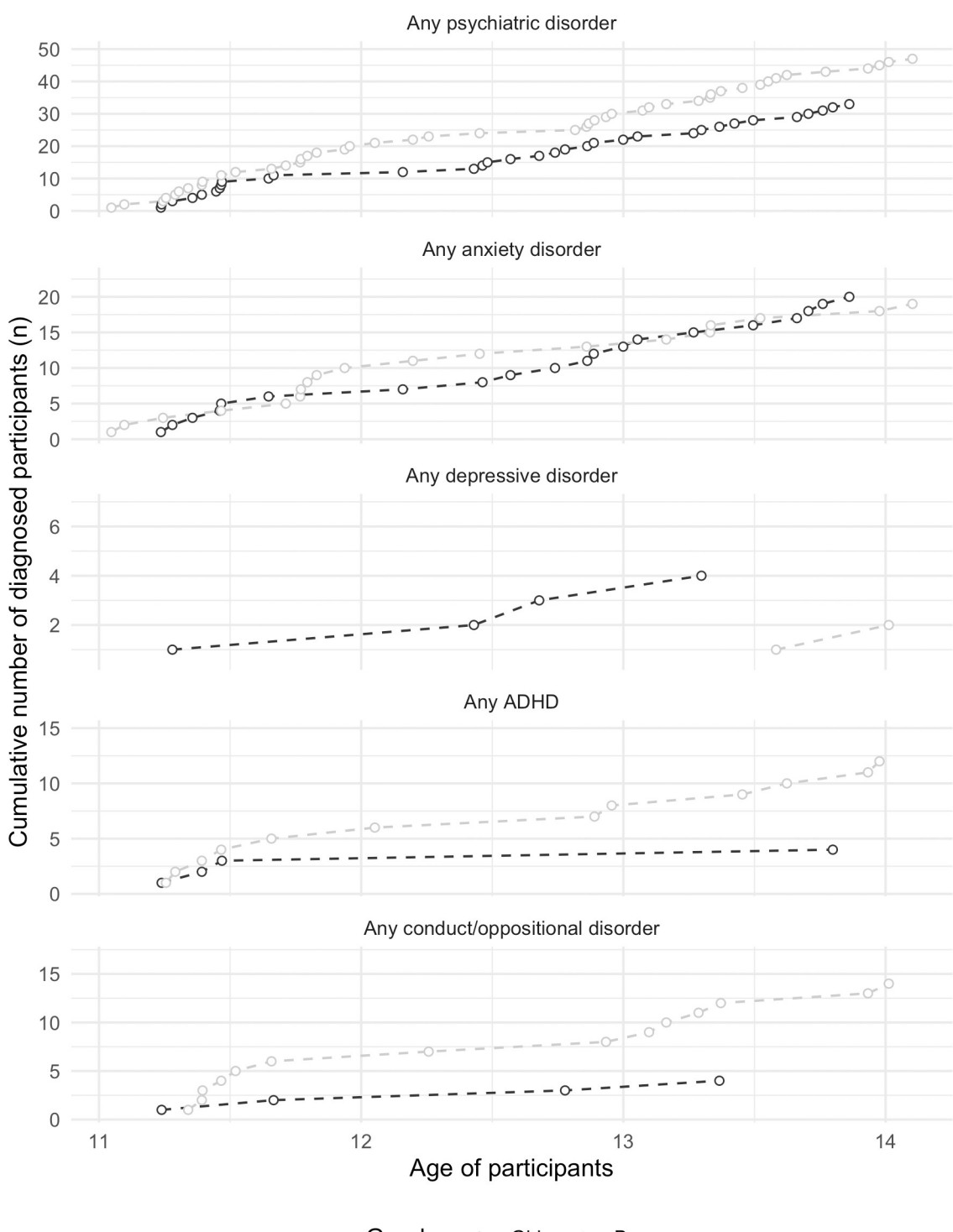

**Fig 3. Cumulative number of diagnosed participants by age.** Observed frequencies. The points represent the cumulative number of participants with a diagnosis at each age. The scales on the y-axis are free as the cumulative frequencies vary by psychiatric disorder.

studies [11]. The upper confidence interval for the estimate is 1.3% lower than the lowest interval estimate reported for any psychiatric disorder in the recent review by Polanczyk et al [11]. For specific disorder categories, the interval estimates for anxiety disorders (2.50–5.85),

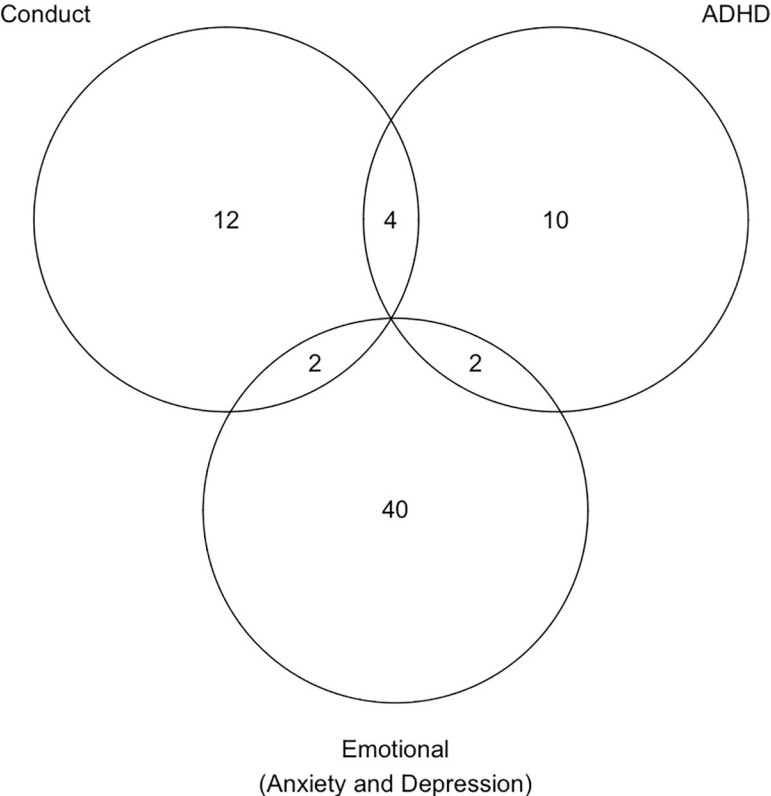

**Fig 4. Venn diagram illustrating overlap between main diagnostic categories.** There were no children with comorbidity across all diagnostic categories. Anxiety and depressive disorders were collapsed into the category Emotional disorders for the purpose of producing this figure.

depressive disorders (0.16–2.58) and ADHD (0.76–3.10) overlap with interval estimates previously reported [11]. The prevalence of disruptive disorders (0.76–3.10), and specifically oppositional defiant disorder (0.19–2.56) are also close to intervals previously reported [11]. For conduct disorder (0.07–0.81), the upper interval estimate obtained in the current study is about 1% lower than the lowest interval estimate reported by Polanczyk et al [11].

The relatively low prevalence rates presented here could be related to methodological issues such as the sampling procedure and choice of diagnostic instrument. Participants were recruited through schools, a sampling frame which may exclude high-risk groups (e.g., sick or disabled children). Still, the DAWBA was completed on-line by parents which may lower the risk of non-participation by high-risk children. The choice of diagnostic instrument has been found to strongly influence prevalence estimates [11]. The time frame of the DAWBA is 'current', which may produce lower prevalence rates compared to other instruments with a wider time frame (e.g., the time frame for CAPA is 3 months, 6 months for DISC, and 1 year for CIDI). In a study where different instruments (the DISC, CAPA, and DAWBA) were used to make diagnoses on the same sample of 1,200 children and adolescents, the DAWBA made only half the number of DSM-IV diagnoses compared to the other instruments [44].

These lower prevalence rates could also be indicative of real and substantial differences in the prevalence of psychiatric disorders between countries. Other studies from Scandinavia have reported low prevalence of parent reported symptoms of mental disorders when using checklists [15–17], but also when diagnostic instruments were used [10,18]. A study comparing externalizing disorders in the current sample with a sample of children from the UK also

found lower scores for externalizing disorders, and this difference was not explained by reporting style [45]. The reasons for the lower prevalence estimates are not obvious. It may be related to the relatively low poverty rates and relatively small income inequalities in Norway [46], to the relatively high education levels in the population [47], or to low unemployment rates [47]. More positive attitudes towards people with mental health disorders [48] may also influence whether certain behaviours are "normalized" or reported as problematic symptoms. However, in a study where rates between the UK and Norway were compared, no evidence for such reporting bias could be demonstrated for DAWBA diagnoses [45]. Further, universal health care and school mental health services may be related to early intervention and prevention of mental health problems [49], and the cultural context may create different exposure to risk and protective factors for the population [50].

## Gender and age differences

The prevalence of any psychiatric disorders was about as frequent among boys and girls. There was considerable overlap in the confidence intervals for the prevalence estimates for males and females, and the lack of strong gender differences was confirmed by gender being a non-significant predictor for all the major categories of psychiatric disorders in the regression analyses. The gender-specific prevalence intervals estimated in the current study are within intervals previously reported for anxiety disorders and ADHD [13,21]. For girls, the prevalence estimates for depression, conduct disorder and oppositional defiant disorder in the current study are also in line with previous reports, but the prevalence of conduct disorder and depressive disorders for boys are lower [13,21].

The gender specific associations were mostly in the expected directions as indicated by the point prevalence estimates, but the imprecision related to low number of diagnosed participants produced wide and overlapping confidence intervals. Anxiety disorders was higher for girls (girls [g]: boys [b] ratio: 1.3:1) as indicated by the point prevalence. We expected a female preponderance of anxiety disorders at this age, in line with several previous studies [13,51,52]. The point prevalence of depressive disorders was also higher for girls compared to boys (g:b ratio: 13:1). A higher rate of depressive disorders among girls was as predicted, as the age of the current sample corresponds well with the age when more females than males develop depressive disorders [53–55]. The point prevalence of disruptive disorders was higher for boys than for girls (b:g ratio: 1.8:1). This result is in line with previous reviews were disruptive disorders have been found to be 1.6–3.1 times more common among males than females [21,22]. An unexpected finding was that the point prevalence of ADHD was relatively similar for boys and girls, even slightly more common among girls (g:b ratio: 1.1:1). This is contrast to results reported in a previous review, where the prevalence of ADHD was estimated to be 3.2–4.7 times higher for male compared to female 5-19-year-olds. In the unweighted analyses in the current study, the male to female ratio was more in line with previous studies (b:g ratio 3.2:1 for ADHD), suggesting there were some unanticipated effects associated with the population weighting for this disorder.

In general, there were gender-specific effects related to applying population weights. With the exception of anxiety disorders, population weighting was associated with an increase in prevalence for girls, and a reduction in prevalence for boys. Analyses of these affects suggested that for some disorders the correlations with parental education levels went in the opposite direction for boys and girls. We also found a stronger association between low parental education levels and prevalence of some disorders for girls compared to boys. However, the results remained essentially unchanged with regards to gender differences, as the prevalence intervals overlapped also in the unweighted data. Still, these gender-specific associations between

disorders and parental education levels may be specific to this sample and should be taken into consideration when interpreting the results from the current study.

The prevalence of psychiatric disorders was not significantly higher at any particular age. Age was a non-significant predictor in the regression analyses, and the prevalence intervals of any psychiatric disorders were overlapping for the youngest and oldest participants. Although the point prevalence was highest for the younger participants, it was highly variable within each age interval. This is consistent with previous studies [13] but may also reflect the few participants diagnosed within some age-intervals. There was evidence of psychiatric disorders accumulating with increasing age, but there was no consistent pattern in the prevalence of disorders in each age-group.

## Strengths and limitations

Among the strengths of the current study are the relatively large sample size, the use of a complete clinician-rated semi-structured interview for establishing psychiatric diagnoses, and the use of post-stratification weights to counter bias related to selective participation.

Some limitations should be taken into account when interpreting the results from this study. One limitation relates to the statistical uncertainty related to few participants being diagnosed which may lead to sample specific arrangements of gender, disorders and parental education levels. Weighting may have amplified these combinations, in particular affecting prevalence estimates for disorders with few participants.

Furthermore, reweighing for non-response implicitly implies a conceptualization of missing data mechanisms [56]. The auxiliary variables available in the current study were parental education levels, so weights by design only address non-response related to parental education levels. Other variables, either where population information is unavailable (e.g., screening scores for mental health problems) or that were not included in the current survey (e.g., household income), may also be predictive of non-response [28].

Another limitation may be related to the lack of self-reported information from children. However, studies have not found additional informants to produce significantly different prevalence rates [11].

Finally, the data for this study was gathered in 2006, and may therefore not reflect current prevalence rates in the population. Our results, however, are consistent with studies of Norwegian children that have been conducted both earlier [10] and later [18], suggesting stability in the prevalence of mental health disorders in the child population. Temporal stability in prevalence was also suggested by a recent review [11], where year of data collection (ranging from 1985 to 2012) was non-significantly associated with prevalence rates of mental health disorders. Still, more recent changes, such as increases in child poverty [57] and the common use of social media with potential implications for mental health [58] underscores the need for updated prevalence estimates.

## Conclusion

In line with previous studies from Scandinavia, the prevalence of psychiatric disorders was found to be relatively low in this sample of Norwegian children. Given the high worldwide prevalence and burden of disease of mental disorders, these findings raise important scientific questions for future research into why populations in some societies show lower rates of mental health problems than others. The findings also suggest that collapsing prevalence estimates across larger regions (such as e.g., Europe), may hide geographical variation which in turn could inform hypotheses about the causes of this variation. The findings from the current

study also underscores the utility of epidemiological studies in providing precise information needed for planning and scaling preventive and interventive services.

## Supporting information

**S1 Table. Prevalence of DSM-IV disorders among 10-14-year-old participants in the Bergen Child Study.**
(DOCX)

**S2 Table. Prevalence of DSM-IV disorders among 10-14-year-old participants in the Bergen Child Study, by gender.**
(DOCX)

## Author Contributions

**Conceptualization:** Tormod Bøe.

**Data curation:** Tormod Bøe, Einar Røshol Heiervang, Kjell Morten Stormark, Astri J. Lundervold, Mari Hysing.

**Formal analysis:** Tormod Bøe.

**Funding acquisition:** Einar Røshol Heiervang, Kjell Morten Stormark, Astri J. Lundervold.

**Investigation:** Einar Røshol Heiervang, Mari Hysing.

**Methodology:** Tormod Bøe, Einar Røshol Heiervang, Kjell Morten Stormark, Astri J. Lundervold, Mari Hysing.

**Project administration:** Tormod Bøe, Einar Røshol Heiervang, Kjell Morten Stormark, Astri J. Lundervold, Mari Hysing.

**Resources:** Kjell Morten Stormark, Astri J. Lundervold.

**Writing – original draft:** Tormod Bøe.

**Writing – review & editing:** Einar Røshol Heiervang, Kjell Morten Stormark, Astri J. Lundervold, Mari Hysing.

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
