## [Decision Letter · Decision Letter 0]

23 Oct 2020

PONE-D-20-25358

Prevalence of Psychiatric Disorders in Norwegian 10-14-Year-Olds: Results from a cross-sectional study

PLOS ONE

Dear Dr. Tormod Boe

Thank you for submitting your manuscript to PLOS ONE. After careful consideration, we feel that it has merit but does not fully meet PLOS ONE’s publication criteria as it currently stands. Therefore, we invite you to submit a revised version of the manuscript that addresses the points raised during the review process.

We look forward to receiving your revised manuscript.

Kind regards,

Ehsan U Syed, MD

Academic Editor

PLOS ONE

Journal Requirements:

2.Thank you for including your ethics statement:  "The study was approved by the Regional Committee for Medical Research Ethics in Western Norway (approval number 062-06) and by the National Data Inspectorate.".   

Please provide additional details regarding participant consent. In the ethics statement in the Methods and online submission information, please ensure that you have specified (1) whether consent was informed and (2) what type you obtained (for instance, written or verbal, and if verbal, how it was documented and witnessed). If your study included minors, state whether you obtained consent from parents or guardians. If the need for consent was waived by the ethics committee, please include this information.

5.We note that you have indicated that data from this study are available upon request. PLOS only allows data to be available upon request if there are legal or ethical restrictions on sharing data publicly. For information on unacceptable data access restrictions, please see http://journals.plos.org/plosone/s/data-availability#loc-unacceptable-data-access-restrictions.

Reviewers' comments:

Reviewer's Responses to Questions

**Comments to the Author**

1. Is the manuscript technically sound, and do the data support the conclusions?

Reviewer #1: Partly

2. Has the statistical analysis been performed appropriately and rigorously? 

Reviewer #1: I Don't Know

3. Have the authors made all data underlying the findings in their manuscript fully available?

Reviewer #1: No

4. Is the manuscript presented in an intelligible fashion and written in standard English?

Reviewer #1: No

5. Review Comments to the Author

Reviewer #1: In this paper "Prevalence of Psychiatric Disorders in Norwegian 10-14 Year-Olds". In this paper, The aim of this study to find prevalence of psychiatric disorders in a Norwegian population-based sample of early adolescent boys and girl. Authors have highlighted limitation of official clinical and administrative registries. Authors also reported prevalence of psychiatric disorder in same cohort between age 7-9 years old in the previous study. I have following concerns about this study.

Method section:

One of the major limitation of this study only 24% completed the full assessment (DAWDA), while this population base study - but only 1/4 completed the assessment and also who completed have more parents education. Authors already reported that about lower socioeconomic status are under-represented as limitation of administrative registry. So how this sample is more representative?

2nd paragraph, authors report that teachers data from the screening phase is not included. However, in abstract and next paragraph (DAWBA was administered through a web-based system, and parents and teachers provided information), and psychiatric assessment section: authors report teacher report was included.

What the rationale of dividing the groups in so close age range? [10.4,11.7), [11.7,12.2), [12.2,12.8), [12.8,13.4) and [13.4,14.2].

Discussion section:

Authors reports low prevalence related to methodological issues as sampling procedure and choice of diagnostic instrument. If we have same issues with this study - how it is better and adding to what we already know?

Authors also reports that low prevalence may be related to DAWDA time frame "current". However, ADHD is chronic psychiatric disorder and anxiety could be also pervasive in childhood ; so how it is different CAPA, DISC or CIDI. Current time frame in DAWDA is also very vague. Authors themselves highlighted the limitation of DAWBA (....only half the number of DSM-IV diagnoses compared to the other instruments [44]). Knowing that this scale have limitation = not sure how this study is representative of population based prevalence of psychiatric disorder.

Minor issues :

2nd paragraph (discussion gender and age difference): Anxiety disorders was higher for girls for anxiety disorders (anxiety disorder twice in a same sentence).

6. PLOS authors have the option to publish the peer review history of their article (what does this mean?). If published, this will include your full peer review and any attached files.

Reviewer #1: No

---

## [Author Response · Author response to Decision Letter 0]

10 Feb 2021

To Editor-in-chief, PLOS ONE.

We would like to thank you for the opportunity to revise our manuscript. Below you will find our responses to the reviewer’s concerns, our responses in italics. We have uploaded a marked copy of our manuscript, as well as an unmarked version to the submission system, accompanying this rebuttal letter.

We would also like to thank the academic editor for extending the deadline for resubmission of our manuscript.

Yours sincerely, 

Tormod Bøe, PhD

On behalf of all authors. 

We have now updated the manuscript according to the PLOS ONE style requirements. 

2.Thank you for including your ethics statement: "The study was approved by the Regional Committee for Medical Research Ethics in Western Norway (approval number 062-06) and by the National Data Inspectorate.".

Please provide additional details regarding participant consent. In the ethics statement in the Methods and online submission information, please ensure that you have specified (1) whether consent was informed and (2) what type you obtained (for instance, written or verbal, and if verbal, how it was documented and witnessed). If your study included minors, state whether you obtained consent from parents or guardians. If the need for consent was waived by the ethics committee, please include this information.

(…)

We have now updated the ethics statement in the manuscript, and in the “Ethics statement” field of the submission form. The revised statement now reads: 

"The study was approved by the Regional Committee for Medical Research Ethics in Western Norway (approval number 062-06) and by the National Data Inspectorate. Participants gave written informed consent and parents and guardians consented on behalf of their children".

We apologize for this error; titles are now identical in the submission form and in the manuscript. 

The list of author affiliations has been updated.

5.We note that you have indicated that data from this study are available upon request. PLOS only allows data to be available upon request if there are legal or ethical restrictions on sharing data publicly. For information on unacceptable data access restrictions, please see http://journals.plos.org/plosone/s/data-availability#loc-unacceptable-data-access-restrictions.

We have updated the Data availability statement to detail the legal restrictions regarding sharing of the data set and also provide this information in the cover letter. The data availability statement now reads:

“Data cannot be shared publicly due to privacy restrictions in accordance with the ethical approval for the Bergen Child Study. Norwegian Health research legislation and the Norwegian Ethics committees require explicit consent from participants in order to transfer health research data outside of Norway. In this specific case, ethics approval is also contingent on storing the research data on secure storage facilities located in our research institution. Data are from the Norwegian Bergen Child Study, owned by NORCE Norwegian Research Centre. The authors did not have special access privileges to data from the survey. Data are available from the Bergen Child Study (contact via bib@norceresearch.no) for researchers who meet the criteria for access to confidential data.”

Reviewers' comments:

Reviewer #1: In this paper "Prevalence of Psychiatric Disorders in Norwegian 10-14 Year-Olds". In this paper, The aim of this study to find prevalence of psychiatric disorders in a Norwegian population-based sample of early adolescent boys and girl. Authors have highlighted limitation of official clinical and administrative registries. Authors also reported prevalence of psychiatric disorder in same cohort between age 7-9 years old in the previous study. I have following concerns about this study.

Method section:

One of the major limitations of this study only 24% completed the full assessment (DAWDA), while this population base study - but only 1/4 completed the assessment and also who completed have more parents education. Authors already reported that about lower socioeconomic status are under-represented as limitation of administrative registry. So how this sample is more representative?

In the authors’ opinions, the strength of the current study is the use of weighting to counter the selective participation by higher educated participants commonly seen in previous literature. By using weights, we use the information about those not participating to estimate what the prevalence would have been if all participants had completed the clinical interview. 

2nd paragraph, authors report that teacher’s data from the screening phase is not included. However, in abstract and next paragraph (DAWBA was administered through a web-based system, and parents and teachers provided information), and psychiatric assessment section: authors report teacher report was included.

This is correct. Teacher data was not included in the screening phase (see Figure 1), but it was included in the psychiatric assessment phase. In the current study, information from the screening phase is used for weighting (parental education levels, reported by parents) and to indicate representativeness of the sample participating in the psychiatric assessment to the full sample (results described under own heading on page 5). 

What the rationale of dividing the groups in so close age range? [10.4,11.7), [11.7,12.2), [12.2,12.8), [12.8,13.4) and [13.4,14.2].

The rationale for dividing the age groups in this way were to provide a detailed perspective on the prevalence of different disorders across this narrow age span. As discussed, (p. 6) the prevalence was highly variable depending on how age was categorized, and we therefore also report the main effects of (uncategorized) age (and its interaction with gender) in the manuscript (pp. 6-7).

Discussion section:

Authors reports low prevalence related to methodological issues as sampling procedure and choice of diagnostic instrument. If we have same issues with this study - how it is better and adding to what we already know?

As detailed above, the weighting procedure used in the current study is aimed at alleviating problems with sampling procedures from previous studies. Regarding the use of DAWBA, this is merely a point to consider when interpreting the results from the current study, and when comparing and contrasting this study to previous studies where different instruments may have been used. 

Authors also reports that low prevalence may be related to DAWDA time frame "current". However, ADHD is chronic psychiatric disorder and anxiety could be also pervasive in childhood ; so how it is different CAPA, DISC or CIDI. Current time frame in DAWDA is also very vague. Authors themselves highlighted the limitation of DAWBA (....only half the number of DSM-IV diagnoses compared to the other instruments [44]). Knowing that this scale have limitation = not sure how this study is representative of population based prevalence of psychiatric disorder.

There is no particular threat to the validity of the current study in light of having used the DAWBA as a diagnostic tool. The different time frame may, however, be useful to be aware of when comparing and contrasting prevalence estimates from different studies where other instruments and time frames may have been used. There is no reason to expect that the current estimates are more or less valid than previous estimates. 

Minor issues :

2nd paragraph (discussion gender and age difference): Anxiety disorders was higher for girls for anxiety disorders (anxiety disorder twice in a same sentence). 

Thank you for noticing this error. We have now corrected this sentence in the manuscript (page 9).

---

## [Editor Report · Decision Letter 1]

8 Mar 2021

Prevalence of Psychiatric Disorders in Norwegian 10-14-Year-Olds: Results from a cross-sectional study

PONE-D-20-25358R1

Dear Dr. Boe

We’re pleased to inform you that your manuscript has been judged scientifically suitable for publication and will be formally accepted for publication once it meets all outstanding technical requirements.

Kind regards,

Ehsan U Syed, MD

Academic Editor

PLOS ONE
---

## [Editor Report · Acceptance letter]

10 Mar 2021

PONE-D-20-25358R1 

Prevalence of Psychiatric Disorders in Norwegian 10-14-Year-Olds: Results from a cross-sectional study 

Dear Dr. Bøe:

I'm pleased to inform you that your manuscript has been deemed suitable for publication in PLOS ONE. Congratulations! Your manuscript is now with our production department. 

Kind regards, 

on behalf of

Dr. Ehsan U Syed 

Academic Editor

PLOS ONE